# The Use of Qualitative Methods to Guide the Development of the Border Resilience Scale in a Participatory Research Study

**DOI:** 10.3390/ijerph20095703

**Published:** 2023-05-01

**Authors:** Maia Ingram, Karina R. Dueñas, Idolina Castro, Luis Vázquez, Rebecca M. Crocker, Emily K. Larson, Jill Guernsey de Zapien, Emma Torres, Scott C. Carvajal

**Affiliations:** 1Arizona Prevention Research Center, Department of Health Promotion Sciences, Zuckerman College of Public Health, University of Arizona, Tucson, AZ 85724, USA; krduenas@arizona.edu (K.R.D.); carvajal@email.arizona.edu (S.C.C.); 2Campesinos Sin Fronteras, Somerton, AZ 85350, USA; 3Cancer Center, College of Medicine, University of Arizona, Tucson, AZ 85724, USA; rcrocker@arizona.edu

**Keywords:** U.S.-Mexico Border, community resilience, scale development, qualitative methods, participatory research, Mexican-origin populations, health outcomes, emotional health

## Abstract

U.S.-Mexico border residents experience pervasive social and ecological stressors that contribute to a high burden of chronic disease. However, the border region is primarily composed of high-density Mexican-origin neighborhoods, a characteristic that is most commonly health-promoting. Understanding factors that contribute to border stress and resilience is essential to informing the effective design of community-level health promotion strategies. La Vida en La Frontera is a mixed-methods, participatory study designed to understand factors that may contribute to border resilience in San Luis, Arizona. The study’s initial qualitative phase included interviews with 30 Mexican-origin adults exploring community perceptions of the border environment, cross-border ties, and health-related concepts. Border residents described the border as a Mexican enclave characterized by individuals with a common language and shared cultural values and perspectives. Positive characteristics related to living in proximity to Mexico included close extended family relationships, access to Mexican food and products, and access to more affordable health care and other services. Based on these findings, we co-designed the 9-item Border Resilience Scale that measures agreement with the psychosocial benefits of these border attributes. Pilot data with 60 residents suggest there are positive sociocultural attributes associated with living in border communities. Further research should test if they mitigate environmental stressors and contribute to a health-promoting environment for residents.

## 1. Introduction

Mexican-origin individuals are the largest Latino/a sub-group in the United States, and more than half of these individuals live in the four US-Mexico Border states [1]. Mexican-origin persons also account for the greatest percentage of the immigrant population, with approximately 11.4 million Mexican immigrants living in the U.S. today [2]. Social determinants that negatively affect the health of Mexican-origin persons are magnified on the US-Mexico border, where several factors coalesce to make the region a challenging environment. Border residents are twice as likely to live in poverty, attend fewer years of formal education, and experience higher rates of unemployment than the population of any individual U.S. state [3]. These conditions translate directly into social and economic contexts that influence health behaviors and health status. A border prevalence study found that nearly two-thirds of Mexican-origin adults had diabetes or pre-diabetes and that half of those with diabetes were undiagnosed [4]. Further, those diagnosed with diabetes by a medical provider still had issues accessing and maintaining care. A qualitative study of decision-making among persons with chronic disease found that beyond the cost of health care, economic stress contributed to the denial of illness and delay in seeking health services until the illness was advanced, which was then exacerbated by poor interpersonal interactions with health care providers [5]. Further, immigration laws focused on border security have drastically increased the militarization of border communities over the past two decades normalizing the discrimination, racial profiling, and mistreatment of border residents [6]. Border residents may also experience stress related to the dominance of images and stories in national news and social media that emphasize criminality, violence, and illegality on the border [7]. Constant exposure to negative depictions of the border may cause residents to internalize feelings of separateness or “otherness” from the mainstream, excluding them from the right to social resources and protections.

Despite the pervasive stressors facing Mexican-origin persons in border communities, there are aspects of Mexican culture that may offer health protection among border-dwelling individuals. As a group, Latino/as enjoy a longer life expectancy than other populations and have a lower risk of mortality from most diseases [8]. Greater life expectancy has also been documented specifically among Mexican-origin individuals living in border communities [9]. Ruiz et al. (2016) propose the sociocultural resilience model (SRM) as a means to conceptualize attributes of Latino/a culture that may be health-promoting [10]. The SRM posits that cultural values associated with strong family ties and interpersonal relationships work in tandem with strong reciprocal social networks to increase social integration to buffer stress and ultimately improve health outcomes [11]. The border environment provides an important context for exploring these and other sociocultural processes inherent in Mexican-origin communities that may promote resilience and mediate stressful conditions.

In this article, we describe the qualitative development of a Border Resilience Scale (BRS) that seeks to characterize and measure attributes of border environments that may increase resilience and be conducive to health. The BRS was derived from themes identified through analysis of in-depth interviews exploring stress and resilience among border residents in southern Yuma County, Arizona. The BRS presents an opportunity to explore the relationship between positive perceptions of border life and health behavior and health outcomes. While linguistically assigning gender, we use Latino/a as an all-encompassing term inclusive of all people of Latin American origin. Latino/a is widely used and culturally accepted within the community we are studying. Mexican-origin more directly describes our participants as this more closely represents their identity.

### 1.1. Potential Sources of Resilience in Mexican-Origin Border Communities

While media and research reports tend to portray the southern US borderlands in a negative light, emphasizing high crime rates, uncontrolled migration, and “spillover” violence from Mexico, in fact, there is evidence that the border region may afford a health-promoting environment [12]. In the realm of physical and emotional health outcomes, research suggests that border residents may benefit from the intersection of cultural and immigrant identities. Shared immigration experiences, along with proximity to their country of origin, may produce a unique border enclave for border residents in which common cultural characteristics coalesce to create a health-promoting environment that counteracts environmental challenges [9]. Border families also reflect generations of immigration experiences which may translate to stronger family identity, support, and mutual care [13]. The shared sociocultural identity of Mexican-origin individuals alone may confer a health advantage on border residents. This cultural advantage may not be limited to those living in border communities. A study of high-density Mexican neighborhoods in the southwest US also found a “barrio advantage” in health outcomes [14]. However, given the overwhelming economic and social challenges facing border residents, there is value in understanding specific attributes of border communities that may contribute to resilience. The extent to which border residents hold positive perceptions of their neighborhoods and communities may have implications for their health and wellbeing.

Cross-border social and economic ties between immigrants and their home countries constitute another potential source of resilience unique to border-dwelling individuals who possess the legal immigration status or dual citizenship to promote cross-border engagement. While there is extensive research describing risk and resilience stemming from the immigrant experience [15], the majority of these studies focus on enclaves within the U.S. and immigrant connections to their home countries [16]. There is less research about the influence of these ties within the context of immigrants who live in physical proximity to Mexico. In terms of sociocultural resilience, cross-border ties may provide an ongoing source of emotional support rooted in familial and ethnic identity [15]. This is compelling because the immigration research on social ties assumes the stress of separation from family, whereas many Mexican-origin individuals with legal status who live on the border cross routinely to see family and access a broad range of goods and services [5].

### 1.2. The Role of Qualitative Methods in Scale Development

There are numerous quantitative instruments that seek to measure resilience, but none have been developed specifically within the cultural context of Latino/a populations or within specific Latino/a communities such as the U.S.-Mexico border [17]. Qualitative approaches often constitute the exploratory phase of creating a quantitative measure of a phenomenon under study, in this case, border resilience. Observation, interviews, and focus groups inform or provide insight into an already identified area of interest or gap in current research or reveal perspectives and experiences not yet identified by researchers. While it is customary for survey or scale development to emerge from qualitative data, detailed descriptions of the process for generating these measures often go unreported [18]. Rowan and Wulf (2007) emphasize the value of describing methods of scale development, not only in identifying the themes or constructs that are measured through quantitative methods but also in providing a description of the context in which it was developed, thereby making transparent the assumptions of those conducting the research and indicating other perspectives that might not have been captured through the process. Moreover, a detailed description of how qualitative methods can complement multiple stages of scale development can be useful to other researchers. In their example, Rowan and Wulf describe the initial stage of developing a scale to measure affiliation with a 12-step program, emphasizing how they first identified the need to better understand and describe the phenomenon among users of the program, explaining the theoretical underpinnings of their conceptual understanding and finally detailing the process of conducting, analyzing and reflecting on the interview process [18].

The qualitative phase of measure development also serves to identify issues that are important to the population under study [19]. In developing a scale on youth anxiety in managing autism spectrum disorder, Bears et al. (2016) engaged parents of children with autism in initial focus groups to better understand the experiences of their children. Over the course of the study, they modified the focus group questions to address new themes as they emerged. They also requested that the parents review existing scales for language and relevance to their children. The authors discuss the challenges inherent in using parent perspectives on manifestations of anxiety in their children. Despite this shortcoming, the authors were able to use parents’ examples of their children’s behavior that could serve as indicators of anxiety in the scale. Finally, they reviewed the 8-item instrument with parents who participated in the focus groups and consulted with outside experts [20].

Qualitative methods in scale development can also bring attention to understudied health issues and populations. While there are various measures of sexual functioning, for example, the need to understand the phenomenon among cancer survivors is specific in terms of the effects of cancer treatment. After a review of the literature that identified the gap, Flynn et al. (2011) conducted focus groups to explore physical intimacy and sexuality among cancer survivors. The focus group guide was initially based on the literature review and expertise of the research group, and again the researchers revised focus group questions as new themes were identified in the initial groups. The analysis elucidated the importance of sexual intimacy to cancer survivors and identified various themes related to fertility and sexual attractiveness experienced by the population, making evident the need to revise existing instruments to capture these cancer-specific experiences [21].

Finally, the qualitative exploratory phase of scale development can explicitly address a theorized social phenomenon and its relationship to health outcomes. Wade and Harper (2021) studied racialized sexual discrimination experienced by young men of color in digital spaces, first grounding a conceptual model in the literature and then presenting the model and asking for feedback on related questions in the context of focus groups. They describe their analysis as a hybrid inductive-deductive approach in which ideas presented in the literature are validated and further explored qualitatively. The study allowed the researchers to deepen their understanding of the proposed constructs, which then informed scale development [22].

In the current study, La Vida en La Frontera (Life on the Border), community and academic partners draw upon these qualitative methods to develop and pilot the Border Resilience Scale (BRS). The BRS aims to address the specific gap in our understanding of the relationship between border life and community resilience as a first step in understanding its relationship to health outcomes.

## 2. Materials and Methods

La Vida en la Frontera is a community-based participatory research study led by academic researchers at the Arizona Prevention Research Center (AzPRC) and community researchers at Campesinos Sin Fronteras (CSF), a grassroots organization founded and led by community health workers (CHWs) that has been addressing the health of farmworker communities in Southern Arizona for three decades. The partnership was initiated in the 1980s, when the now CSF Director participated in a maternal and child health study that established the CHW workforce in the region [23]. The partnership evolved over time to address chronic disease, with mental and emotional health a central focus of disease prevention and management [24]. The partners are committed to participatory decision-making and community benefit in all phases of research and emphasize ongoing reciprocal and mutual learning processes [25].

The SRM is an assets-based explanatory model for understanding resilience within the context of significant economic and social stressors. The premise of the La Vida en La Frontera study is that: (1) stress is positively associated with chronic disease, but that (2) protective factors posited by the SRM will have direct and moderating effects on biological stress and chronic disease risk. As such, the study included a qualitative explanatory phase designed to elucidate the potential protective factors articulated by the SRM, followed by a longitudinal study of the identified factors and their relationship to health outcomes among a cohort of border residents. We developed the BRS within the context of the qualitative phase of the study (see Duenas et al. 2022 for a detailed description of the study protocol) [26].

### 2.1. In-Depth Interviews

The qualitative phase of the study included in-depth interviews with residents of two border communities in Arizona and one community across the border in Sonora, Mexico. To develop the interview guide, our team engaged in extensive discussion about stress, resilience, and community-level characteristics that might influence individual perceptions and experiences. The final interview guide was thus grounded in the lived experience and border identity of the CHW research partners. For example, we made assumptions that respondents would have positive perspectives on border life and that these attributes would be connected to cultural values associated with Mexican culture, such as proximity to family. We also drew on the SRM constructs to explore how cultural values and social ties might operate specifically in the border environment.

The partners developed the interview questions simultaneously in Spanish and English, working collaboratively to resolve the meanings across the translations, and created a bilingual instrument that was regionally and culturally appropriate [27]. The concept of “resilience” in English, for example, was not adequately captured through a direct Spanish translation as “resiliencia.” The final interview thus did not include the specific term to explore resilience but rather explored sources of well-being and strength (fuentes de bienestar y fortaleza), asking questions about how interviewees cope with stress and overcome stressful situations (superar el estrés y seguir adelente). CSF CHWs recruited and scheduled interview participants using a convenience sample of their existing client networks, as well as circulating brief announcements on their local radio station, television station, and newspaper. Community and academic researchers trained in qualitative data methods conducted the interviews. While the COVID-19 surge in 2020 necessitated that we conduct interviews over Zoom phone rather than in person, we found that the pandemic-induced isolation encouraged intimate sharing despite the remote format. The initial interviews led us to revise the interview guide to acknowledge COVID-19 stress, particularly related to its impact on family, as well as how it impacted participants’ experiences of life on the border. The 45 to 90 min interviews were audio recorded and then professionally transcribed in the language in which they were conducted.

### 2.2. Data Analysis

The qualitative analysis reflected an emergent design or a flexible approach in which new information was incorporated across all stages of the study. After a workshop-type discussion on ways to conduct qualitative analyses, we agreed that three members of the research team would code these data, sharing each stage of the process with CSF partners for input, feedback, and interpretation. Our thematic analysis utilized both a deductive approach based on the SRM and the concepts under study, as well as an inductive approach that allowed us to identify common experiences or perspectives raised by participants [28]. Following an initial close reading of several interviews, we identified four broad categories that included ‘the border, ‘stress,’ ‘resilience,’ and ‘health.’ Data related to the development of the BRS and presented in this article were coded under the category ‘the border.’ The academic researchers provided a high-level summary of their perceptions of these data, noting responses that we found unexpected or surprising. These summaries, which included data excerpts, allowed the CSF CHWs to engage directly with these data, providing context for the findings, revising interpretations, and recommending additional exploration.

We then created a detailed codebook with themes and subthemes for each category, collaboratively defining each code, which included illustrative examples from the interviews. Three academic researchers coded a subset of the interviews in Dedoose (www.dedoose.com, accessed on 19 April 2022) so that each interview was double-coded. We then read through the initial analysis resolving differences and clarifying definitions. Discussion between academic and community partners confirmed the commonalities of our perceptions as well as identified additional themes. Two community researchers reviewed the themes, subthemes, and illustrative quotes and provided written interpretations, which were then integrated into the analysis.

## 3. Results

### 3.1. In-Depth Interviews

Thirty individuals participated in the interviews, 17 female and 13 male, ranging between 23 and 60 years of age. All but three individuals were born in Mexico, and all but one reported having family currently living in Mexico. While four respondents crossed the border “rarely” or “not very often” or 7–8 times a year, the other 26 respondents crossed it daily, weekly, or monthly. Approximately one-third of respondents worked in agriculture, another third worked in daycare or preschool education, and the remainder either worked in business/retail, security, or maintenance or were not working.

We identified four related yet distinct themes that described perceptions of the border context that might be conducive to resilience and potentially health-promoting: ‘border life,’ ‘Mexican enclave,’ ‘border crossing,’ and ‘binational connections’ (Table 1). Subthemes displayed in Table 1 summarize attributes under each theme that were expressed across several interviews with illustrative quotes. Focused primarily on the U.S. side, the theme ‘border life’ described positive characteristics or attributes of daily living in a border community. In characterizing their community as both busy and peaceful, respondents described these attributes as being Latino/a or Hispanic. Other culturally grounded attributes included an appreciation of food and a preference for self-care to treat illness over seeking clinical care. Living on the U.S. side of the border was specifically perceived as safe compared to the Mexican side, with more opportunities for education, employment, and economic mobility. Perceptions of safety in the U.S. compared to Mexico were a predominant theme in the interviews and are further explored in Crocker et al. (2022) [29].

Respondents also described their community as a Mexican enclave in which most residents are from Mexico and thus share common experiences and cultural references. Mexican heritage and roots were expressed as a common language that was not English, as well as in shared behaviors. Perhaps the most notable aspect of this theme was the description of the immigration experience, the shared stories through which respondents described their motivations, the opportunities they sought, the challenges they confronted, and even the details of “arreglando los papeles” or filing immigration paperwork.

The third theme, ‘binational connections,’ emphasizes the importance of geographical proximity to Mexico, a characteristic distinct from other Mexican enclaves located in the interior of the U.S. Respondents described the essential contribution to their emotional health in being able to fulfill family responsibilities by caring for aging parents and grandparents, as well as participating in activities with extended family members, such as birthdays, weddings, and baptisms. One respondent explained that about one-fourth of people living on the U.S. side also have houses on the Mexican side and customarily spend the weekend there, creating long lines at the border on Friday afternoons.

The theme “border crossing” described the frequency and motivations for enduring long lines and border enforcement to visit Mexico regularly. Practical reasons for crossing included accessing more affordable health care and medications, including behavioral and mental health services, veterinary services, and shopping for food and specialty items only available or cheaper in Mexico. Respondents also attached a deep emotional benefit to being able to visit their loved ones, visit specific restaurants and purchase Mexican products to take back to the U.S. Notably; several respondents described their connections to specific places that they traveled to when crossing to Mexico, a specific recreational site or beach where they can “clear their heads”. Table 1 summarizes these main subthemes under each theme and provides illustrative quotes.

### 3.2. Border Resilience Scale Development

During the discussion about theinterview data, it became clear that there were several community-level (rather than individual-level) characteristics of border life that could contribute to a shared and positive identity. Since existing instruments generally focus on the individual rather than community attributes of resilience, the research team saw a need for developing a resilience scale specific to community-level attributes within the context of the US-Mexico border region [30]. The themes in our data described several characteristics of border life that could contribute to a shared and positive identity, which in turn could translate to more resilient emotional health. One of the academic researchers drafted the questions, which were then refined through discussion. In an iterative process, the academic researchers cross-checked each question with the themes and subthemes in Table 1. As demonstrated in Table 2, some of the scale items tap into more than one theme. The final BRS addresses three overarching topics: (1) the migration experience as it relates to a pathway to personal and familial safety and economic security; (2) regular access to Mexico to connect with family and cultural identity, as well as to access goods, services, and recreational opportunities; and (3) the tangible and emotional benefits of living in the US and close to the US-Mexico border. The BRS questions are presented in Table 2 and mapped to the themes coded under the border category.

### 3.3. Pilot Survey

To pilot the BRS, we incorporated the questions into a longitudinal survey that is part of Aim 2 in the parent La Vida en La Frontera study designed to examine the interaction of chronic disease risk longitudinally, social determinants of health, psychosocial factors, inflammatory biomarkers, and clinical outcomes. In the parent study, CSF community health workers recruit participants through door-to-door randomized community-based sampling in two U.S. border communities. Eligible individuals complete a longitudinal survey in the home setting with an interview-style approach [26]. With the exception of question six, the nine-item BRS follows a 4-point Likert scale from 1 to 4 (1 = Completely disagree, 2 = Disagree, 3 = Agree, 4 = Completely Agree). Question six measures the frequency of travel across the border into Mexico and employs a skip pattern, whereas if a participant indicates that they do not cross the border, questions seven through nine are not asked. The instrument is scored by adding the response across all questions with the Likert scale.

Data from sixty individuals (n = 60) who were recruited through randomized door-to-door neighborhood outreach into the cohort study was utilized to complete a preliminary analysis of the instrument. Descriptive statistics were used to calculate demographic characteristics and BRS item frequencies using R Statistical Software, R Markdown in RStudio, Version 4.2.1. The majority of the pilot participants were female (76.6%), with a mean age of 54.6 years. This pilot sample’s age is normally distributed with a skewness of −0.27 and a kurtosis of −0.06. Almost all were born in Mexico (91.7%) and had parents (28.3%), adult children (50%), and children under 18 (41.7%) residing in Mexico (Table 3). Table 4 shows the frequency (%), mean score, and standard deviation for each item (and scale total) on the BRS with the exclusion of question 6, which does not follow a Likert Scale. The total mean score for the BRS is μ = 29.5 (SD = 2.9), and each item has a mean score above μ = 3. The majority of participants responded with “Agree” or “Strongly Agree” to all of the questions on the Border Resilience Scale. In addition, 70% indicated that they had crossed the border between the US and Mexico. These responses provide initial evidence that the BRS scale captures the perceived benefits of living in the border region among Mexican-origin individuals and that they perceive these benefits as improving their quality of life.

Notably, 60% of participants “Strongly Agree” that they feel safer living on the US side of the border than they would living in Mexico. These results are congruent with thesqualitative data that describe increased drug violence on the Mexican side of the border and an increasing sense of risk to personal safety [29]. Further, the level of agreement with this question is stronger than the other questions included in the BRS, suggesting that safety is currently a predominant factor when assessing the benefits of border residence.

## 4. Discussion

As an academic concept, resilience has roots in various disciplines. While definitions vary by discipline and often by the unit of analysis dominant within them, a common theme is individual resilience as part of a response to external stressors. In the realm of psychological research, resilience focuses on understanding personal characteristics that enable individuals to recover from adverse events [31]. Community resilience, as defined in a World Health Organization Evidence Synthesis Report (2020), “is the ability of a community to adapt and thrive in response to external stressors” [32]. Community resilience assumes that communities can adapt to external threats and uncertainty in a time of rapid change and is compelling in offering a lens through which to conceptualize collective rather than individual responses to adversity [33]. The WHO recommends community-engaged approaches in identifying community strengths, and a CBPR approach was essential to this study to consider resilience from individual, social, and community perspectives. Community and academic partners reflected upon the assets and challenges of the local context, seeking to understand and measure resilience, with the goal of identifying community and cultural strengths protective toward health outcomes. Within this participatory approach, we identified the need for a scale that focused on aspects of community-level resilience unique to border environments.

The border is a rich environment for the study of community resilience. The southern U.S. border is typically economically and ecologically vulnerable, and most communities are within medically underserved areas, as identified in government reports and databases. Those living in this region are often victims of policies that are generally hostile to people from Mexico and the Northern Triangle, despite the nation’s history as a country of immigrants [34]. The increased militarization and tightening of border security over the past several decades have caused tremendous stress in the daily lives of residents, who are subject to federal law that allows border officials to stop and search individuals within 100 miles of the border [6]. The persistent delays and procedural obstacles that impede border crossing are burdensome for border residents with aging parents and other loved ones across the international boundary line [35]. The geopolitical reality also creates crisis points for residents, such as the constant influx of asylum seekers and the closing of the border during the peak of the COVID-19 pandemic. Border communities consistently respond to these crises by creating humanitarian aid infrastructure powered by volunteers and community organizations [36], perhaps further evidence of their affiliation to a border identity with distinct personal and moral obligations.

Our research also made it clear that living on the border comes with increasing levels of stress that need to be managed. The traditional border literature describes the U.S.-Mexico border as an imaginary boundary [37], but an anti-immigrant sentiment and the physical construction of a steel border wall have ruptured this identity, complicating the experience of border residents [38,39,40]. Border residents are aware of the increased risk of injury and death to those who are driven to cross illegally by desperation or fear of organized crime in their countries of origin. U.S. and Mexican drug policy has also driven a sharp contrast in the levels of violence on either side of the border. Many who preferred to live in Mexico while taking advantage of work and educational opportunities across the line now feel compelled to live in the U.S., expressing their appreciation of the calm and peaceful life that is customarily attributed to US rural and suburban communities [29].

Many people immigrate to the United States from Mexico, however, interviewees expressed a preference for staying on the border because of their comfort with the culture and daily life, including being able to communicate in Spanish, not only between residents but also with government officials and service providers. While unquestionably a Mexican enclave, border residents are heavily influenced by both U.S. and Mexican norms, fashioning ways of thinking about and approaching life that could only be described as binational. The idea of moving into the interior of either country provokes a level of anxiety among residents that surpasses the challenges of navigating the border.

There are attributes that we identified in our study that may not translate to other border communities. The sister cities of San Luis, Arizona, and San Luis Rio, Colorado, are relatively small compared to San Diego and Tijuana, or El Paso and Juárez, for example. In this community, it may be easier to cross back and forth through the port of entry despite the long lines, both in vehicles and pedestrian access. Our study makes it clear that visiting family, whether deported children or a fiancé or a grandma who lives alone, is an emotional and cultural necessity and the strongest motivator for border crossing in this community. Thus, residents of Southern Yuma County incorporate the wait into their concept of the trip, adapting to the long lines and finding relief in the Ranchera music playing on the local radio station. It is notable that in discussing border connections, interviewees did not reference the growing potential for digital technologies to help them maintain relationships but rather focused on physical proximity and face-to-face interactions.

Other characteristics of border life identified in our study, such as accessing medical care and medications in Mexico, have been identified in numerous other studies [5,41,42]. Viewed from the perspective of resilience in a region with the highest rates of uninsured in the U.S. [43], the interviewees described several adaptive responses. First, people put off seeking care as long as possible. Second, they go to Mexico for quicker and less expensive care. Finally, when facing life-threatening illnesses, they seek care in the U.S. under the belief that there is advanced life-saving technology.

La Vida en La Frontera initially approached the study of resilience as a sociocultural construct, and there is an overlap between the SRM and the BRS in measuring the importance of social networks and cultural values. There is a potential benefit, however, in considering the BRS as measuring an explicit form of community resilience because of its congruence with Latino/a collective culture and, more specifically, with Mexican-origin populations [44]. Health-related community resilience is typically assessed by measuring the social determinants of health and the community environment [32], and it is important to consider the contribution of border resilience within that framework. While the BRS measures aspects of the border community that may contribute to resilience, resilience is a highly complex construct that can be measured differently across populations and environments. The BRS does not measure all aspects of resilience characterized by an individual living on the border, and a “low” score on this scale does not indicate that the individual does not have qualities of resilience. Moving forward, the challenge is to determine if the shared community identity translates into the ability to leverage cultural strengths and communal resources that help individuals mitigate stress and sustain emotional and physical well-being.

Our study provides an important example of the importance of qualitative research in exploring and expounding on complex concepts such as community resilience. This qualitative phase of our parent study was essential to unearthing cultural phenomena that were not captured in existing measures. The iterative approach common in qualitative analysis guided us in reviewing and synthesizing these data into broad ideas, which we were then able to narrow into measurable perspectives on the advantages of border life. In doing so, we drew on the methods established in other scale development efforts [18,19,20,21,22]. With SRM as our conceptual model, we used in-depth interviews with border residents to explore perspectives on social ties and cultural values expressed in the model. Data gathered in the interviews and our iterative analysis of these data made it clear that there was a need for a quantitative measure of perceptions of border life as a form of community resilience that may have a positive influence on emotional and physical health. The participatory approach was important to ensuring that community researchers were involved in interpreting these data from the perspective of their experience living on both sides of the U.S.-Mexico border, and we refined the questions through extensive discussion. In detailing the construction and piloting of the BRS, it is our intent to document this first step to measure border resilience within a specific community context, recognizing that there is a need for input and validation from other border communities. The next steps include implementing the BRS within the larger La Vida en La Frontera cohort and evaluating its psychometric and predictive properties. Psychometric properties of the BRS will be assessed using data from the complete participant population and standard analyses to measure the reliability and validity of the instrument as it compares to other measures of resilience and psychosocial processes measured in the study. Predictive properties of the BRS will be assessed using regression models in later study analyses. Additionally, future work should include a further qualitative investigation into perspectives on the scale with community members.

One limitation of our study is the lack of a common definition for community resilience, which makes it difficult to fit the concept of border resilience into a health resilience framework. Proponents of community resilience suggest that this ambiguity allows community members to reflect on their own communities and identify those aspects that might be health-promoting [32]. However, an agreed-upon conceptual framework for community resilience would facilitate measuring the pathways between positive perceptions of one’s community and health outcomes. A related issue is that the BRS is one small piece of a complex web of interactions between individuals and their environments, and drawing connections between the BRS and health outcomes may be overly simple. Further, other border communities or studies may identify additional or different attributes of border resilience that should be considered. Another limitation is that the BRS is a reflection of findings from border residents who had lived in their community for a number of years and had the legal status to cross the border regularly. Additionally, three-fourths of respondents were female, and we cannot determine if males would respond to the scale in a similar way. However, in our interviews, we did not note differences in the ways that men and women described their perceptions of and relationship to the border. In recognizing these challenges, we felt that the strength of these qualitative data in describing positive perceptions of border life warranted a specific focus on this aspect of resilience. We recommend ongoing qualitative and community-engaged work in moving these concepts forward.

## 5. Conclusions

Border resilience may offer an important form of health protection for Mexican-origin individuals living in the U.S.-Mexico border region, and the present work’s findings may be transferrable to many other immigrant enclaves within the Global North. BRS pilot data suggest that the positive attributes of border life represent a common and affirming border identity that may translate into health benefits. The participatory nature of this qualitative research was essential to exploring community context, as well as conceptualizing how external factors, such as migration patterns, family and social network ties, and border enforcement, might cause border resilience to thrive or disintegrate. Further research is needed to explore the concept of border resilience in other border communities, as well as to ascertain the extent to which border resilience is measurable on an individual level and comparable to other high-density immigrant communities. Additionally, ongoing research is needed to understand better the relationship between border crossing patterns, border resilience, and emotional health.

## Figures and Tables

**Table 1 ijerph-20-05703-t001:** Perceptions of the Border.

Theme 1: Border Life		
Sub Themes	Illustrative Quotes	English Translation
The border is active and busy, while with a small-town feel. The border has its own culture. Behavioral norms change from one side of the border to the other. The U.S. side of the border is perceived as a safe and calm place to live as the Mexican side experiences more drug trafficking, shootings, and kidnappings.	- Teresa: Es que es frontera, todo el tiempo es activa. Pero la mayoría de gente que aquí vivió y aquí creció y se conocen entre todos. - Silvia: Porque siento que aquí son como dos tipos de culturas …O no dos tipos de cultura, si no es una cultura como muy única a otros lugares. - Andrea: Lo que me gustaría es …que la misma educación que tienen aquí en los EEUU fuera la misma educación que cuando ellos visitan México. … cuando crucen la frontera no les quiten el cinturón a los niños, déjenselos, no porque la policía allá no diga nada, o sea, les tienen que quitar el cinturón. - Maria: Bueno ahorita lo que estamos viviendo en la frontera, pero sería mexicana, que ahorita hay mucha delincuencia y eso dicen que es mucho basado en que todos quieren tomar las fronteras. Ahorita hay mucho …—pues algo que nunca se había mirado acá, asesinatos, ya secuestros a la luz del día.	- Teresa: It’s because it’s the border, it’s active all the time. But most people lived and grew up here and know each other. - Silvia: I think there are two kinds of cultures, or not two cultures, but it is a very unique culture from other places. - Andrea: What I would like is… that they practice the same habits in the US as in Mexico…Like, when they cross the border don’t unfasten their seatbelts, leave them on, just because the police there don’t say anything doesn’t mean they should unfasten their seatbelt. - Maria: Well, right now, what we are experiencing on the border, on the Mexican side, right now, there is a lot of crime, and they say that it is because everyone wants to take over the borders. Right now, there is a lot, well, something that has never been seen here, murders and kidnappings in the middle of the day.
Theme 2: Mexican Enclave		
The border population is mostly Latino/a and from Mexico. Culture is expressed through the Spanish language and also through attitudes about health and behaviors. Residents share immigration histories, with many describing the specific challenges that motivated them to migrate to the U.S., as well as the preference to stay close to Mexico.	- Natalia: La gente aquí pues todos casi la mayoría son de México, el vecindario donde yo vivo pues en sí casi todo San Luis, Arizona, todos venimos de México. Es raro—aquí no hay gente como Americana, casi aquí no hay. Hay más Mexicanos. - Dora: nosotros los Latinos tenemos esa costumbre de que si estoy bien nunca me atiendo. Entonces, ya vamos cuando ya en verdad estamos enfermos, muy enfermos. - Natalia: Yo siempre he pensado que nosotros los Mexicanos o la comunidad Hispana siempre somos bien preocupones, siempre estamos preocupados pensando más adelante. - Gloria: Cuando recién mi esposo me inmigro, me arreglo papeles para estar legalmente aquí, nos venimos para acá y aquí nos quedamos. … Gracias a Dios no hemos tenido necesidad de movernos más para adentro de los Estados Unidos.	- Natalia: The people here, well, almost all of them are from Mexico; the neighborhood where I live, well, almost all of San Luis, Arizona itself, we all come from Mexico. It’s weird—there are no Americans here. There are more Mexicans. - Dora: We Latinos have this habit that if we’re not sick, we don’t go to the doctor. So, we only go when we are truly sick, very sick. - Natalia: I have always thought that we Mexicans, or the Hispanic community, we are worriers; we are always worried and thinking ahead. - Gloria: When my husband recently immigrated, I arranged papers to be here legally, we came here, and we stayed here. … Thank God we have not had the need to move further into the United States.
Theme 3: Binational Connections		
Reasons for staying by the border are numerous, the most common being family connections and responsibilities. The ability to visit regularly is essential to emotional health. People were also motivated by their affinity for Mexican culture, and many led a double life with houses and families on both sides of the border.	- Gloria: Pues no sé, que no pudiéramos cruzar. Ay sabe que eso si me estresaría mucho. … muchos de los que vivimos aquí en la frontera, tenemos familia en México. - Anabel: Pienso que el lugar que nos gustó es ahí porque mi mamá tiene familia acá, para México y está pegadita y cuando hacen fiestas, cosas, allá en San Luis más cerca. Y porque como éramos de acá también pues estamos cerquita, siempre extrañamos acá, siempre sale de México. Por eso nos fuimos a vivir a San Luis, Arizona. Pero ahí estamos en San Luis, Arizona, ya de ahí no nos movemos.	- Gloria: Well, I don’t know what we would do if we couldn’t cross. Oh, you know that would stress me out a lot. ….many of us who live here on the border have family in Mexico. - Anabel: I think we liked this place because my mom has family there in Mexico, and it is close, and when they have parties, things, there in San Luis, it’s closer. And because we were from there, we yearn for it. We always miss Mexico. That’s why we went to live in San Luis, Arizona. But, there we are in San Luis, Arizona, and we won’t move from there.
Theme 4: Border Crossing		
In general, residents are accustomed to crossing the border with a range of frequency. While reasons for crossing are clearly tangible (work, health care, cheaper goods), there is also a deep cultural attachment to culture (food, interpersonal relationships, recreation) and place.	- Daniel: Ella no tiene a nadie más que la ayude y por eso vamos. Con la abuelita de mi esposa también. Y pues eso, más que nada para eso, para visitar a nuestros –nuestros seres queridos. - Norma: Más que nada, mi mama le gusta mucho el mandado de allá, de México, las tortillas y el jabón, cositas así. Y si vamos seguido a las farmacias, va y se medica ahí, a las hierberías. - Jose: Mi esposa, que el quesito, que las cosas de México, que dice ella que están más buenas. Entonces, siempre traíamos para hacer tamales, hacer esto, lo otro. - Luis: ¡Pues si! Si cruzo la frontera porque voy al doctor en San Luis México, a San Luis Rio Colorado…Ya tengo dos perros, y a mis dos perras las llevo a Mexicali al veterinario. Este es que esta mas barato allá [se ríe]. - Beatriz: Los viajes también de recreación como ir a visitar lugares para esparcir uno la mente, para sacar ese estrés también nos gustaba mucho… para Ensenada.	- Daniel: She doesn’t have anyone else to help her, and that’s why we go. With my wife’s grandmother too. And well, that, more than anything for that, to visit our –our loved ones. - Norma: More than anything, my mom really likes shopping in Mexico, the tortillas and this soap, little things like that. And we often go to the pharmacies, to get medications, to the herb shops. - Jose: My wife, the cheese, the things from Mexico, which she says are better. So, we always brought stuff to make tamales, to make this or that. - Luis: Well, yes! If I cross the border because I go to the doctor in San Luis, Mexico, in San Luis Rio Colorado...I already have two dogs, and I take my two dogs to the vet in Mexicali. This one is cheaper there [he laughs]. - Beatriz: Also recreational trips like going to visit places to relax one’s mind, to get rid of that stress we also liked a lot... for Ensenada.

**Table 2 ijerph-20-05703-t002:** Border Resistances Scale Mapped to Interview Themes.

BRS Question	Spanish	Themes
Q1. I feel at home living close the border because almost everyone in my community is also of Mexican origin.	Me siento a gusto viviendo cerca de la frontera porque casi todos en mi comunidad también son de origen Mexicano.	Mexican enclave Border Life
Q2. I feel at home living close to border because almost everyone in my community speaks Spanish.	Me siento a gusto viviendo cerca de la frontera porque casi todos en mi comunidad hablan Español.	Mexican enclave
Q3. I feel at home living close to the border because it is easy to find Mexican foods and other products.	Me siento a gusto viviendo cerca de la frontera porque es fácil encontrar comidas y otros productos Mexicanos.	Mexican enclave Border crossing
Q4. I feel at home living close to the border because a lot of my family lives nearby or visits me.	Me siento a gusto viviendo cerca de la frontera porque mucha de mi familia vive cerca o me visita.	Binational connections Border crossing Mexican enclave
Q5. I feel safer living on the US side of the border than I would feel living in Mexico.	Me siento más seguro viviendo en el lado Estadounidense de la frontera que viviendo en Mexico.	Border Life
Q6. How often, if at all, would you cross back and forth between the United States and Mexico?	Antes de la pandemia de COVID-19, ¿con que frecuencia cruzaría de un lado a otro entre los Estados Unidos y Mexico?	Border crossing Binational connections
Q7. It helps me that I can cross the border to see family and friends in Mexico.	Me ayuda que pueda cruzar la frontera para ver a familiares y amigos en Mexico	Binational connections Border crossing
Q8. It helps me that I can cross the border to connect with my culture in Mexico.	Me ayuda que puedo cruzar la frontera para conectarme con mi cultura en México.	Border crossing Binational connections
Q9. It helps me that I can cross the border to get medical services, health supplies, or other goods and services in Mexico.	Me ayuda que puedo cruzar la frontera para obtener servicios médicos, suministros de salud o otros bienes y servicios en México.	Border crossing

**Table 3 ijerph-20-05703-t003:** BRS Pilot Participant Demographics. n = 60.

Demographics	Mean (Range)	Standard Deviation
Age (years)	54.6 (19.6–83.2)	13.7
Years living in neighborhood	17 (1–50)	12
Years living in city	21.6 (1–50)	12
Years living in the U.S.	29.2 (18–48)	12.4
	Percent	Number
Female	76.6	46
Born in Mexico	91.7	55
Parents residing in Mexico	28.3	17
Children (under 18) residing in Mexico	11.7	7
Adult children residing in Mexico	18.3	11

**Table 4 ijerph-20-05703-t004:** Border Resilience Scale Pilot Responses. n = 60.

Total Border Resilience Scale Score					Mean (Range)	Standard Deviation
	Strongly agree # (%)	Agree # (%)	Disagree # (%)	Strongly disagree # (%)	25–9 (−2.0–32.0)	2.9
**Q1.** I feel at home living close the border because almost everyone in my community is also of Mexican origin.	33.3	66.7	0.0	0.0	−0.3 (3.0–4.0)	0.5
**Q2.** I feel at home living close to border because almost everyone in my community speaks Spanish.	25.0	75.0	0.0	0.0	3.3 (3.0–4.0)	0.4
**Q3.** I feel at home living close to the border because it is easy to find Mexican foods and other products.	25.0	71.7	1.7	1.7	3.2 (1.0–4.0)	0.6
**Q4.** I feel at home living close to the border because a lot of my family lives nearby or visits me	30.0	61.7	8.3	0.0	3.2 (2.0–4.0)	0.6
**Q5.** I feel safer living on the US side of the border than I would feel living in Mexico.	60.0	38.3	1.7	0.0	3.6 (2.0–4.0)	0.5
**Q6.** How often, if at all, do you cross back and forth between the United States and Mexico?	**Cross the border** **n = 42**					
**Q7.** It helps me that I can cross the border to see family and friends in Mexico.	18.3	80.0	1.7	0.0	3.2 (2.0–4.0)	0.4
**Q8.** It helps me that I can cross the border to connect with my culture in Mexico.	13.3	75.0	8.3	0.0	3.1 (2.0–4.0)	0.5
**Q9.** It helps me that I can cross the border to get medical services, health supplies, or other goods and services in Mexico.	21.7	61.7	10.0	3.3	3.1 (1.0–4.0)	0.7

## Data Availability

Not applicable.

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
