# Peer review of "The Use of Qualitative Methods to Guide the Development of the Border Resilience Scale in a Participatory Research Study"

_ijerph, 2023, doi:10.3390/ijerph20095703_

Round 1
Reviewer 1 Report
See attached

Author Response
Thank you for your review. We are grateful for your overall positive response and your comments to improve the manuscript. Below are our responses.
Comment: It may be worth noting that the border connections the authors describe are NOT based in any digital technologies—social media, phones, and others. It’s based on old fashioned proximity and face to face meetings with family, friends, and co-ethnics (combined with feeling safer across that border at night!). May be cliché in the public health field to say that digital technologies cannot substitute for actual interpersonal relationships, but it’s an important issue nonetheless.
Response: We agree that it is interesting that the interviewees did not bring up ways that had used technology to save them time waiting in border lines. Asking about this more specifically might be a good area for future research. We added your comment to the discussion section:
It is notable that in discussing border connections, interviewees did not reference the growing potential for digital technologies to help them maintain relationships, but rather focused on physical proximity and face-to-face interactions.
Comment: I have no particular expertise in QM, but it appears to me that they have used very appropriate methods to develop the BRS themes. I would encourage the researchers to take those themes back to the people they interviewed (or others) to see if the themes make sense and to get people to elaborate on why those themes matter. Understanding the social processes behind these themes as well as possible is critical to understanding how the border may translate into better health outcomes. They would very likely be nuanced at a minimum and likely improved by doing so. I do NOT make this suggestion as a requisite for publishing this MS, of course.
Response: We complelety agree with you and we will integrate some form of this activity in the third stage of the research project, thank you. We also incorporate this intention under our recommended next steps.
Additionally, future work should include further qualitative investigation into perspectives on the scale with community members
Comment: Page 10, Line 318: Rephase “70% indicated that they ever cross the border. My brain filled in “never cross the border” and I had to re-read it.
Response: Good point, we have removed the word “ever”
Comment: It seems that, despite the problems at the border, people report many positive aspects of living there. In fact on page 12 lines 369-377 the authors make several uncited claims that people prefer the border and moving to the interior would be anxiety-provoking. (Are these arguments from interviews “many people…… prefer to stay on the border.”) However, I wonder if many people are actually “trapped” or immobile because of their obligations to family and friends. Is the border region where they actually chose or is it where they have to live because of these obligations? I suspect the authors would say that it’s a positive choice for the people they interviewed, but I suspect that border residents elsewhere may prefer to live in a Mexican enclave or community “in the interior” away from border violence and chaos if it weren’t for family obligations at the border.
Response: We appreciate the comment and overall reflection. It would probably be a good addition to ask if border residents would move if they did not have obligations. For the purposes of the current study, we have clarified that the interviewees expressed a preference to staying near the border. This now reads: "interviewees expressed a preference for staying on the border" and this sentiment was also expressed in Table 1 under the theme "border enclave"
Reviewer 2 Report
This is a well-written study. I do not have many comments. Overall, this is a well-done study—well-written, with a solid sample, clear methodological approach, and creative process of scale production. Consider more literature on issues faced by immigrants before the resiliency factors, a bit more on what you might do to further validate the scale. And provide recruitment procedures and the reliability of the measure you derived. This was an enjoyable study to read.
Format/writing: I have very few comments here, as the document is quite well-written. My comments correspond with line numbers. Note that I have almost no comments here, as the writing is excellent.
· 29: At several places I would have expected a comma between two main clauses joined by coordinating conjunctions (and, but, or, so)
· English translation: “I have always thought that us Mexicans..”à or “…that we Mexicans… are worriers”
· “Cuándo recién um mi esposo…” I’m not sure what the “um” is here. If it is a verbal pause, you might leave it out (as none of the other answers have verbal pauses in them).
· “Pero ahí estamos…ya de ahí no nos movemos”—double-check audio. “Aquí” and “ahí” are close in sound. Since the speaker seems to be speaking from Arizona (“acá”) they would more likely refer to Arizona as “here” than “there.”
· “…las tortillas y este el jabón.” The translation says “this” soap. My guess is that the speaker used a filled pause—Spanish uses “este” for a filled pause rather than “uh” in English. “el jabón” translates to “the soap” which makes sense in the context (the tortillas and the soap). I would probably eliminate “este” from the Spanish as a filled pause.
· “…siempre traíamos para hacer tamales…” The English translation adds “to make corn” which does not appear in the Spanish.
· Table 2: In some places, “It helps me that I can cross…” is translated with the present indicative “que puedo,” and in other cases, with the present subjective “que pueda.” Choose one and translate it the same way each time.
· 322: Consider making left column flush left rather than centered, for readability. But this is your and publishers’ choice.
· 422, 434: There are a couple of dangling modifiers [As a participatory study…; As a community development approach).
Substantive comments:
· I greatly enjoyed the overall focus of the article. It is nice to see research on the positive aspects of immigrant communities especially, as you note, in view of the common rhetoric and research that highlights only negative aspects of the immigrant experience.
· 47: If anything, you could do more to cover research on stressors. I like that you included immigration laws. Be sure to include the “rhetorical” climate of political messages, social media, and legacy media that “Others” immigrants in general and Mexican immigrants especially, before you turn to the resilience factors (btw, of course I would say this—communication is my academic field).
· 99-100: Make the connection between the 1.1 and 1.2 clearer. How does one lead to the other. Section 1:2 gives clear understanding of the usefulness of qualitative research to derive quantitative measures. I could provide more citations from my own discipline to this effect, but they would be superfluous to what you already have.
· 168ff: I was a little unclear about the “hypotheses” of the SRM. Since it is a model, perhaps these are “propositions” (for example, if one is using Blalock’s traditional model of “theory” building, with the hypotheses being the statements to be tested in a specific study).
· Method explanation, overall, is solid. The one thing I don’t recall seeing is anything about recruitment.
· 202, 230: 30 interviews, good variety (approximates idea of “maximum variation” sample that Lincoln & Guba and others propose); with 45-90 min interviews. This is a rich data set.
· 200: The changing of the interview guide is absolutely in line with qual researchers like Lincoln & Guba (Naturalistic Inquiry, 1985), Strauss & Corbin (grounded theory) and others. Perhaps note that you are using “emergent design” to round out your theoretical development to gain as many concepts as you can for the survey you are development. (I wouldn’t use the phrase grounded theory” in this study—and you don’t).
· Great detail throughout method would meet the standards of those who talk about the credibility of qual research,. Including coding, creating codebook, etc.
· Table 1 is a little difficult to read. Work with journal perhaps to put lines between columns or between rows. As I read the findings, I felt the original explanation of the categories seemed very close, like there was overlap in some of the categories. The exemplars did help clear this up a little.
· 292: I really liked the idea of creating a scale from the qual, and this seems useful for future research. My question above about the distinction between the categories in the results comes up again here, as some of the items explicitly tap more than one item. Perhaps address this explicitly (i.e., that some of the items tap more than one theme, as the themes are interwoven). This might have implications for testing scale properties.
· 300: I’m guessing that these interviews were “standardized interviews,” asking each participant the questions in the same way, offering the closed-ended options?
· 307: “were recruited”: how?
· Table 4 is clearly presented. Did you calculate the alpha reliability of the measure? Will next steps include verifying the “construct validity” of the scale (for example, through confirmatory factor analysis) or other scale-verification techniques?
· Discussion is well-developed and thoughtful.
· 414: “The iterative approach inherent in qualitative research”—whether it is inherent or not depends on the nature of the qualitative research. I would say “common’ rather than “inherent”
· 413: “…were not captured in existing measures.” To strengthen the importance of your findings, be more specific: What were you able to unearth that was not in previous measures (give citations for some of those measures).
· On limitations, consider sample limitations: you note the median age, but there is a wide age range in the sample with a fairly large standard deviation. This might not be that strong of a limitation.
Author Response
Response: Thank you for your review. We are grateful for your overall positive response and your comments to improve the manuscript. Below are our responses.
Comment: Consider more literature on issues faced by immigrants before the resiliency factors.
Response: We have added the following with three citations at the end of the first paragraph in the introduction. We also have more extensive discussion of stressors in the discussion.
Border residents may also experience stress related to the dominance of images and stories in national news and social media that emphasize criminality, violence and illegality on the border. 7 Constant exposure to negative depictions of the border may cause residents to internalize feelings of separateness or “otherness” from the mainstream excluded them from the right to social resources and protections.
Comment: a bit more on what you might do to further validate the scale…. and the reliability of the measure you derived
Response: In the discussion we added the following:
Psychometric properties of the BRS will be assessed using data from the complete participant population and standard analyses to measure the reliability and validity of the instrument as it compares to other measures of resilience and psychosocial processes measured in the study. Predictive properties of the BRS will be assessed using regression models in later study analyses.
Comment: And provide recruitment procedures.
Response: We added the following to the first paragraph of the methods section:
CSF CHWs recruited and scheduled interview participants using a convenience sample of their existing client networks, as well as circulating brief announcements on their local radio station, television station and newspaper.
Comment: At several places I would have expected a comma between two main clauses joined by coordinating conjunctions (and, but, or, so)
Response: We reviewed the article for these conjunctions and added commas.
Comment: on English translation: “I have always thought that us Mexicans..”à or “…that we Mexicans… are worriers”
Response: We have made that edit.
Comment: “Cuándo recién um mi esposo…” I’m not sure what the “um” is here. If it is a verbal pause, you might leave it out (as none of the other answers have verbal pauses in them).
Response: We have made that edit.
Comment: “Pero ahí estamos…ya de ahí no nos movemos”—double-check audio. “Aquí” and “ahí” are close in sound. Since the speaker seems to be speaking from Arizona (“acá”) they would more likely refer to Arizona as “here” than “there.”
Response: The interviewee may have been in Mexico during the interview. We reviewed the translation to make sure that the respondent said ‘there.’
Comment: “…las tortillas y este el jabón.” The translation says “this” soap. My guess is that the speaker used a filled pause—Spanish uses “este” for a filled pause rather than “uh” in English. “el jabón” translates to “the soap” which makes sense in the context (the tortillas and the soap). I would probably eliminate “este” from the Spanish as a filled pause.
Response: We have made that edit.
Comment: “…siempre traíamos para hacer tamales…” The English translation adds “to make corn” which does not appear in the Spanish.
Response: You are right! We removed the reference to corn.
Comment: Table 2: In some places, “It helps me that I can cross…” is translated with the present indicative “que puedo,” and in other cases, with the present subjective “que pueda.” Choose one and translate it the same way each time.
Response: Thank you for this observation, we have edited this.
Comment: 322: Consider making left column flush left rather than centered, for readability. But this is your and publishers’ choice.
Response: Agreed, we also like it flush.
Comment: 422, 434: There are a couple of dangling modifiers [As a participatory study…; As a community development approach).
Response: We have rewritten these sentences.
Comment: 47: If anything, you could do more to cover research on stressors. I like that you included immigration laws. Be sure to include the “rhetorical” climate of political messages, social media, and legacy media that “Others” immigrants in general and Mexican immigrants especially, before you turn to the resilience factors (btw, of course I would say this—communication is my academic field).
Response: We have added the following with three citations at the end of the first paragraph in the introduction. We also feel we expanded on these issues in the discussion section.
Border residents may also experience stress related to the dominance of images and stories in national news and social media that emphasize criminality, violence and illegality on the border. 7 Constant exposure to negative depictions of the border may cause residents to internalize feelings of separateness or “otherness” from the mainstream excluded them from the right to social resources and protections.
Comment: 99-100: Make the connection between the 1.1 and 1.2 clearer. How does one lead to the other. Section 1:2 gives clear understanding of the usefulness of qualitative research to derive quantitative measures..
Response: We have clarified the need for better understanding of resilience as a community and border construct at the beginning of section 1.2 which now reads
There are numerous quantitative instruments that seek to measure resilience, but none were developed within the specific cultural context of Latino/a populations or within specific Latino/a communities such as the U.S..-Mexico border.18
Comment: 68ff: I was a little unclear about the “hypotheses” of the SRM. Since it is a model, perhaps these are “propositions” (for example, if one is using Blalock’s traditional model of “theory” building, with the hypotheses being the statements to be tested in a specific study).
Response: We have changed hypotheses to “premise”.
Comment: Method explanation, overall, is solid. The one thing I don’t recall seeing is anything about recruitment.
Response: We have added that information in the first paragraph of the methods:
CSF CHWs recruited and scheduled interview participants using a convenience sample of their existing client networks, as well as circulating brief announcements on their local radio station, television station and newspaper.
Comment: Perhaps note that you are using “emergent design” to round out your theoretical development to gain as many concepts as you can for the survey you are development. (I wouldn’t use the phrase grounded theory” in this study—and you don’t).
Response: Great suggestion. We added this at the beginning of the analysis section:
Qualitative analysis reflected an emergent design, or a flexible approach in which new information is incorporated across all stages of the study.
Comment: Table 1 is a little difficult to read. Work with journal perhaps to put lines between columns or between rows. As I read the findings, I felt the original explanation of the categories seemed very close, like there was overlap in some of the categories. The exemplars did help clear this up a little.
Response: In addition to formatting the table, we have clarified this in two places, first in the results section we added in line 214 a better description of Table 1:
Subthemes in Table 1 summarize attributes under each theme that were expressed across several interviews with illustrative quotes.
Under section 3.3., we rewrote the sentence to say
In an iterative process, the academic researchers cross checked each question with the themes and subthemes in Table 1. As demonstrated in Table 2, some of the scale items tap into more than one theme.
Comment: 292: I really liked the idea of creating a scale from the qual, and this seems useful for future research. My question above about the distinction between the categories in the results comes up again here, as some of the items explicitly tap more than one item. Perhaps address this explicitly (i.e., that some of the items tap more than one theme, as the themes are interwoven). This might have implications for testing scale properties.
Response: As mentioned above, in addition to better describing Table 1, we added the following in the description of creating the scale:
In an iterative process, the academic researchers cross checked each question with the themes and subthemes in Table 1. As demonstrated in Table 2, some of the scale items tap into more than one theme.
Comment: 300: I’m guessing that these interviews were “standardized interviews,” asking each participant the questions in the same way, offering the closed-ended options?
Response: This is correct:
Comment: 307: “were recruited”: how?
Response: This information is in the previous paragraph which reads:
In the parent study, CSF community health workers recruit participants through door- to -door randomized community-based sampling in two U.S. border communities. Eligible individuals complete a longitudinal survey in the home setting with an interview style approach.26
Comment: Table 4 is clearly presented. Did you calculate the alpha reliability of the measure? Will next steps include verifying the “construct validity” of the scale (for example, through confirmatory factor analysis) or other scale-verification techniques?
Response: Thank you for this comment, we have addressed the limitations of the size of this pilot sample and described future steps in the discussion section as:
Psychometric properties of the BRS will be assessed using data from the complete participant population and standard analyses to measure the reliability and validity of the instrument as it compares to other measures of resilience and psychosocial processes measured in the study. Predictive properties of the BRS will be assessed using regression models in later study analyses.
Comment: 414: “The iterative approach inherent in qualitative research”—whether it is inherent or not depends on the nature of the qualitative research. I would say “common’ rather than “inherent”
Response: We have made that edit.
Comment: 413: “…were not captured in existing measures.” To strengthen the importance of your findings, be more specific: What were you able to unearth that was not in previous measures (give citations for some of those measures).
Response: Thank you for this comment, we have included a survey of the literature on resilience measures in Latino/a populations and also the following sentence in response to a similar comment by reviewer 1.
There are numerous quantitative instruments that seek to measure resilience, but few were developed within the cultural context of Latino/a populations or within specific Latinx communities such as the U.S..-Mexico border.18
Comment: On limitations, consider sample limitations: you note the median age, but there is a wide age range in the sample with a fairly large standard deviation. This might not be that strong of a limitation.
Response: We reviewed our data and with respect to age, the sample has a normal curve. We clarified this in the results and also added the limitation in the discussion.
Reviewer 3 Report
I enjoyed the article and learned much from it. The culture of the border communities makes their proximity to the former homeland, as you stipulate, an important source of "resilience". The mediating element here is the importance of family in that culture. There is, I suggest, more that can be unpacked from the specifics of that shared culture, and that might explain the importance of maintaining family connections, and one such more thing is a shared religion. About that, as opposed to a shared language, you say nothing. Maybe it would be helpful if you did.
Minor quibble: "We Latinos", "us Mexicans" (p 7): be consistent. The first use is correct English grammatical usage, the second, not so. Use, please, "we Mexicans".
Author Response
Reviewer 3:
Thank you for your review and we are grateful for the overall positive response to the paper. We respond to your comments below.
Comment: There is, I suggest, more that can be unpacked from the specifics of that shared culture, and that might explain the importance of maintaining family connections, and one such more thing is a shared religion. About that, as opposed to a shared language, you say nothing. Maybe it would be helpful if you did.
Response: In the interviews, the respondents did express the importance of religion and spirituality in addressing stress, however they did not talk about religion in the context specifically of the border or their border identity. Religion and spirituality are very important in studies and measures of resilience and many instruments on resilience include questions on religion, as well as on language. The distinction we found in our interviews was the contextual aspect of the border that we attempted to capture in our questions.
Comment: Minor quibble: "We Latinos", "us Mexicans" (p 7): be consistent. The first use is correct English grammatical usage, the second, not so. Use, please, "we Mexicans".
Response: Thank you for this observation, we have made this correction.
Reviewer 4 Report
This is an important study of a timely topic in this era of immigration debates. The authors have presented an important dilemma -- prior research on the negative consequences of immigration on health, yet they have found a community cum enclave in which shared cultural practices can provide support that promotes good health. Then they developed a mixed-methods approach to assess these factors that arose in qualitative research.
My only suggestion concerns data analysis. You discuss that this population sews "older" -- e.g., average age in 50s. They can also note that their sample skews female and add a sentence about if they think gender may factor into questions of support or tension.
Other than this, well. done. This should be published.
Author Response
Thank you for your review and we are grateful for your positive response. To address you point about the fact that the sample for the pilot study is 75% female we added the following:
This pilot sample’s age is normally distributed with a skewness of -.27 and a kurtosis of -.06.
In the limitations section, we added:
Additionally, three-fourths of respondents were female, and we cannot determine if males would respond to the scale in a similar way. However, in our interviews, we did not note differences in the ways that men and women described their perceptions of and relationship to the border.
Reviewer 5 Report
This is a very important and interesting study; however, I have a few formal comments and comments regarding the theoretical backgroud and clarity as to which residency and before which factors are involved.
1) in the Results section, the demographic analysis should be placed first, followed by the other results;
2) At the beginning of the article, the concept of residuals needs a thorough definition, there is a very extensive literature here; what is residuals for the authors, how do they define them in the context of the research?
3) Since the scale used for the research shows in a highly general way and on the basis of an assessment by n=3-participants of their own subjective, personal feelings, experiences, needs, the positioning of Mexican cultural-ethnic minorities in American contexts, it is not very clear what 'resilience' specifically refers to (negative propaganda or so-called black PR in the media is not everything): whether it is about a subjective sense of alienation, whether it is about some kind of restriction, discrimination, problems with access to health care locally, or perhaps about excessive, compared to 'traditional health customs' ('we go to the doctor when we are already very sick = i.e. in the culture described here, preventive health care, for example, is neglected. preventive health care) pressures and expectations from the public health system in the USA; or is it that in health care facilities, doctors and nurses question some health decisions of the population with a mixed Mexican-US background; in my opinion, resilience is not about a person pushing away all new surroundings, but about self-defence against some unjust pressures, regulations, attacks, claims; a bit lacking in this article are the specific factors that resilience is supposed to protect against; and there is also the question of whether everything new around is automatically 'harmful' and 'bad'. It would have been useful to have a few sentences to extract some dominants from the haze of statements. Perhaps this term 'border resilience' is not quite clear in the context of health care? Once these points have been revised, the article will be suitable for publication.
Author Response
Thank you for your review and thoughtful reflection. We believe that other reviewers had similar comments and believe that our revisions have addressed these issues. Responses to your specific comments are below:
Comment1) in the Results section, the demographic analysis should be placed first, followed by the other results;
Response: The demographics are currently first in the article as mentioned. They read:
Thirty individuals participated in the interviews, 17 women and 13 men, ranging between 23 and 60 years of age. All but three individuals were born in Mexico and all but one reported having family currently living in Mexico. While four respondents crossed the border “rarely” or “not very often” or 7-8 times a year, the other 26 respondents crossed daily, weekly or monthly. Approximately one-third of respondents worked in agriculture, another third worked in daycare or preschool education, and the remainder either worked in business/retail, security, or maintenance or were not working.
Comment 2) At the beginning of the article, the concept of residuals needs a thorough definition, there is a very extensive literature here; what is residuals for the authors, how do they define them in the context of the research?
Response: We believe that the reviewer meant to say that resilience needed a good definition. We specifically avoided an extensive review of the concept of resilience, but rather grounded our study in the sociocultural resilience model, which we defined in the introduction. The concept of resilience emerged from the qualitative analysis, and we hope that we addressed the reviewer’s comment in the discussion section which more closely describes addresses the concept of community resilience and includes more literature which is related to contextual resilience versus individual resilience.
Comment: 3) Since the scale used for the research shows in a highly general way and on the basis of an assessment by n=3-participants of their own subjective, personal feelings, experiences, needs, the positioning of Mexican cultural-ethnic minorities in American contexts, it is not very clear what 'resilience' specifically refers to (negative propaganda or so-called black PR in the media is not everything): whether it is about a subjective sense of alienation, whether it is about some kind of restriction, discrimination, problems with access to health care locally, or perhaps about excessive, compared to 'traditional health customs' ('we go to the doctor when we are already very sick = i.e. in the culture described here, preventive health care, for example, is neglected. preventive health care) pressures and expectations from the public health system in the USA; or is it that in health care facilities, doctors and nurses question some health decisions of the population with a mixed Mexican-US background; in my opinion, resilience is not about a person pushing away all new surroundings, but about self-defence against some unjust pressures, regulations, attacks, claims; a bit lacking in this article are the specific factors that resilience is supposed to protect against; and there is also the question of whether everything new around is automatically 'harmful' and 'bad'. It would have been useful to have a few sentences to extract some dominants from the haze of statements. Perhaps this term 'border resilience' is not quite clear in the context of health care? Once these points have been revised, the article will be suitable for publication.
Response: We very much appreciate the comments of the reviewer. We received similar comments from reviewer one and added these elements into the introduction in the description of the border stressors. These speak to the reference to subjective sense of alienation that might be experienced by border residents:
Border residents may also experience stress related to the dominance of images and stories in national news and social media that emphasize criminality, violence and illegality on the border 7. Constant exposure to negative depictions of the border may cause residents to internalize feelings of separateness from the mainstream that they are excluded from social resources and protections.
With respect to the other comments about border resilience as a construct, we hope that the discussion section with the additional revisions address some of these points. In the discussion we attempt to place the qualitative data and the border scale within the broader context of the border and include a description of the realities of border life that the reviewer is commenting on.